# Investigation of Air Pollutants Related to the Vehicular Exhaust Emissions in the Kathmandu Valley, Nepal

**Yukiko Fukusaki [1,*], Masataka Umehara [1], Yuka Kousa [1], Yoshimi Inomata [1] and Satoshi Nakai [2]**

1   Yokohama Environmental Science Research Institute, Yokohama 221-0024, Japan;
    ma05-umehara@city.yokohama.jp (M.U.); yu01-kousa@city.yokohama.jp (Y.K.);
    yo03-inomata@city.yokohama.jp (Y.I.)
2   Department of Artificial Environment, Graduate School of Environment and Information Sciences, Yokohama
    National University, Yokohama 240-8501, Japan; nakai-satoshi-dc@ynu.ac.jp
*   Correspondence: yu00-fukusaki@city.yokohama.jp; Tel.: +81-45-453-2550

**Abstract:** The Kathmandu Valley, which is surrounded by high hills and mountains, has been plagued by air pollution, especially in winter. We measured the levels of volatile organic compounds, nitrogen dioxide, nitrogen oxides, sulfur dioxide, ammonia, ozone, $PM_{2.5}$, and carbon monoxide in the Kathmandu Valley during the winter to investigate the impact of vehicular emissions and the contribution of gaseous air pollutants to secondary pollutants. The most common gaseous pollutants were discovered to be gasoline components, which were emitted more frequently by engine combustion than gasoline evaporation. Considering the ethylene to acetylene ratio, it was discovered that most vehicles lacked a well-maintained catalyst. Compared to previous studies, it was considered that an increase in the number of gasoline vehicles offset the effect of the measures and exceeded it, increasing the level of air pollutants. Aromatics and alkenes accounted for 66–79% and 43–59% of total ozone formation potential in Koteshwor and Sanepa, respectively. In terms of individual components, it was determined that ethylene, propylene, toluene, and m-xylene all significantly contributed to photochemical ozone production. As those components correlated well with isopentane, which is abundant in gasoline vehicle exhaust, it was determined that gasoline vehicles are the primary source of those components. It was indicated that strategies for regulating gasoline vehicle exhaust emissions are critical for controlling the photochemical smog in the Kathmandu Valley.

**Keywords:** gasoline; diesel; well-maintained catalyst; photochemical ozone production

## 1. Introduction

Kathmandu is the capital city in Nepal, located in the Kathmandu Valley with east–west 25 km and south–north 20 km at an altitude of 1300–1400 m above sea level. Kathmandu Valley has four distinct seasons: pre-monsoon, monsoon, post-monsoon, and winter. The winter season is the most polluted [1,2]. Kathmandu is surrounded by high mountains and hills. This geographical feature restricts horizontal dilution of air pollutants from the valley area, especially when temperatures are low and winds are calm. Regmi et al. [3] established relationships between air pollution potential and local meteorology in the Kathmandu Valley during the winter: the valley wind begins to blow from the west around 12:00, the wind is calm around 19:00, the ground inversion layer develops due to radiant cooling during the night, the ground inversion layer continues to develop until approximately 8:00, the level of air pollutants peaks, and the level of air pollutants gradually declines during the day due to the developing mixing. Thus, the level of air pollutants in the Kathmandu Valley is largely affected by geographical and meteorological conditions.

The population in the Kathmandu Valley is growing rapidly. The valley's population increased from 766,345 in 1981 to 2,510,788 in 2011 and is expected to reach 5,744,964 by 2031 [4]. The number of vehicles is increasing as the population increases. It combines with the geographical and meteorological conditions to aggravate air pollution in the

Kathmandu Valley. In 2017, the number of vehicles increased to ~100,000 compared with ~34,000 in 1990 [4]. Motorcycles have the highest number of registered vehicles. The use of gasoline and diesel has increased significantly in tandem with the growth of vehicles over the years. There are no large industrial facilities in the Kathmandu Valley, such as iron and steel plants, petrochemical complexes, or refineries. Volatile organic compound (VOC) sources in the Kathmandu Valley, excluding vehicles, include brick manufacturing, boilers, waste combustion, domestic, agriculture, construction, and power plants. Diesel is primarily used in boilers and power plants, whereas liquefied petroleum gas (LPG) is primarily used in domestic activities. Other fuels commonly used are coal, wood, and rice husk [4].

Several studies have reported the impact of various sources of air pollution. In a previous study on source apportionment, four primary sources (motor vehicles, soil dust sources, brick kilns, and biomass/garbage burning) were identified, and 95% of the average $PM_{10}$ concentration is attributed to local primary sources such as motor vehicles (31%), soil dust (26%), biomass/garbage burning (23%), and brick kilns (15%) [5]. According to Shakya et al. (2017) [2], the most significant particulate matter (PM) sources in the Kathmandu Valley are vehicle emissions and resuspended dust. It was also stated that the atmospheric polycyclic aromatic hydrocarbons in Kathmandu were primarily caused by the combustion of fossil fuels (including vehicle exhaust and coal combustion) and biomass fuels [1]. Based on the foregoing, it is reasonable to assume that vehicular exhaust emissions contribute significantly to the severe air pollution in the Kathmandu Valley.

The Nepalese government has introduced several policies for controlling air pollution. Table 1 summarizes major policy initiatives targeting vehicular emissions [4]. The government banned new three-wheelers, popularly known as Vikram Tempos, from Kathmandu's streets in 1991; in 2000, it implemented the Nepal Vehicular Mass Emission Standard, which is equivalent to European Emission Standard 1 (Euro 1); in 2017, it imported EURO IV fuel quality, among other things. However, there has been no improvement in air quality as a result of rapid urbanization, rapid population growth, increased vehicle ownership, and increased fuel consumption. Among those who have contributed to this work are Shakya and others. The authors of [6] reported that the median hourly average $PM_{2.5}$ concentration in the vicinity of six major road intersections exceeded 100 $\mu g \cdot m^{-3}$, with the 5 min maxima reaching above 1000 $\mu g \cdot m^{-3}$ in winter 2014, exceeding 25 $\mu g \cdot m^{-3}$ defined as the limit per year by World Health Organization. In addition, Putero et al. [7] also reported that $PM_{2.5}$ presented an average value of 195 $\mu g \cdot m^{-3}$ near the edge of Kathmandu's tourist district.

**Table 1.** Major policy initiatives targeting vehicular emissions in the Kathmandu Valley.

| Year | Policy |
| --- | --- |
| 1991 | Banned diesel three-wheelers registration |
| 1994 | Emission standards for in-use vehicles |
| 1999 | Banned three-wheelers operated by diesel |
| 1999 | Subsidies for electric vehicles |
| 2000 | Nepal Vehicle Mass Emission Standard EURO I |
| 2000 | Stopped two stroke registration |
| 2001 | Announced the ban of 20 year old vehicles, but not implemented |
| 2001 | National Transport Policy |
| 2003 | National Ambient Air Quality Standards |
| 2004 | Two strokes three-wheelers banned from operation |
| 2009 | National indoor air quality standard and implementation guideline |
| 2012 | EURO III standard |
| 2017 | Phase out of 20 year old public transport and goods vehicles |
| 2017 | Import of EURO IV fuel quality |

Air pollutants emitted from vehicles are mainly gaseous pollutants and particle matters, which are emitted from gasoline and diesel vehicles, respectively. Most of the studies related to air pollutants in the Kathmandu Valley focused on PM which was the main air

pollutant [5,6,8–11]. On the contrary, volatile organic compounds (VOCs) are critical precursors, which react with OH radicals in the presence of solar radiation to generate ozone ($O_3$) which is harmful to human health and contributes to global warming. When VOCs are emitted from gasoline vehicles, there are two patterns: the first is fuel evaporation, such as breakthrough or permeation during parking, immediately after stopping, while driving on the road, and so on, and the second is exhaust emissions from tailpipes after fuel combustion. Exhaust gas is composed of both fuel combustion and non-combustion gas. As combustion efficiency varies according to engine performance, the ratio of combustion gas to non-combustion gas varies by vehicle type. Therefore, it is critical to identify the contribution of each fuel and the degree of impact of each vehicular-related emission process. However, few studies on the details of VOCs have been conducted [12,13].

This study focused on gaseous pollutants such as VOCs and explores the contribution of vehicles and the degree of impact on each vehicular-related emission process, as well as identifies the species that largely contributed to the secondary pollutants in the Kathmandu Valley. Vehicle contributions were calculated using the relative impacts of gasoline (GV), diesel (DV), and LPG vehicles (LV). Correlations between combustion and non-combustion gas components, as well as the ethylene to acetylene ratio, were used to evaluate the degree of influence on each vehicular-related emission process. Secondary pollutants were compared using ozone formation potential (OFP) as an indicator. This study provides useful information for planning the strategies of vehicles and/or energy to mitigate air pollution.

## 2. Methods

### 2.1. Sampling

Air samples were collected at two sites in Kathmandu, Nepal: Koteshwor and Sanepa (Figure S1). Koteshwor is an intersection with heavy traffic (Figure S2). Sanepa is a residential neighborhood (Figure S3). Koteshwor was chosen to collect fresh vehicular exhaust, whereas Sanepa was chosen to collect aged air emitted by vehicles over time. The sampling heights at Koteshwor and Sanepa were ~1 and 10 m above the ground, respectively. Sample collections were conducted between 7 and 11 March 2019. The days of 8 and 9 March were holidays. During the sampling period, the weather was pleasant.

The following components were analyzed: nitrogen dioxide ($NO_2$), $NO_x$, sulfur dioxide ($SO_2$), ammonia ($NH_3$), $O_3$, $PM_{2.5}$, carbon monoxide (CO), and VOCs (128 components) including 28 alkanes, 23 alkenes, 20 aromatics, 3 biogenic, 2 aldehydes, 3 ketones, 9 oxygenated and 40 others, with the meteorological parameters (ambient temperature, atmospheric pressure, relative humidity, wind direction, and wind speed). VOCs were collected at 9:00, 11:30, 14:00, and 16:30 on 7 March in Koteshwor, and 10 March in Sanepa, with $PM_{2.5}$ and CO levels and meteorological parameters. On other days, aldehydes and $O_3$ were collected for 24 h. During the sampling period, gaseous pollutants such as $NO_2$, $NO_x$, and $NH_3$ were collected four times every 24 h. $SO_2$ was collected for 4 days.

VOC samples were collected at a flow rate of ~30 mL·min$^{-1}$ for 10 min using Summa-polished stainless steel canisters (Scientific Instrumentation Specialists, Moscow, ID, USA) and four samples were collected at each sampling site. Using BPE-DNPH cartridges (Merck KGaA, Darmstadt, Germany), aldehyde samples were collected at a flow rate of 1 L·min$^{-1}$ using pumps (GSP-400FT; Gastec Inc., Japan) for 2.5 h sampling, 200 mL·min$^{-1}$ for 16.5 and 24 h sampling, respectively. Seven samples were collected at each sampling site.

Passive samplers with the filters (Ogawa Incorporated, Japan) were used to collect $NO_2$, $NO_x$, $SO_2$, and $NH_3$ [14]. The filters for $NO_2$ and $NO_x$ were impregnated with triethanolamine and 2-phenyl-4,4,5,5-tetramethylimidazoline-3-oxide-1-oxyl (PTIO), while the filters for $SO_2$ and $NH_3$ were impregnated with triethanolamine and citric acid, respectively. The passive samplers were placed at two sampling sites and left for ~24 h to collect $NO_2$, $NO_x$, and $NH_3$, and four samples were collected at each sampling site. As the passive sampler for $SO_2$ was left for 4 days, only one sample was collected at each sampling site. The LD-5 equipped with the cyclone device (Sibata Scientific Technology

Ltd., Saitama, Japan) was used to measure $PM_{2.5}$ levels in the range of 0.001–10000 mg·m$^{-3}$, based on the principle of light scattering. The analyzer was calibrated by background measurement using a standard scattering plate before each measurement with an accuracy of $\pm 10\%$. The measurement interval was set to 10 min and the number of data obtained was 49 and 53 at Koteshwor and Sanepa, respectively. ToxiRAE Pro (RAE System Inc., San Jose, CA, USA) with an electrochemical sensor was used to measure CO in the range of 0–500 ppm (measure resolution of 1 ppm). The analyzer was calibrated using AutoRAE 2 Test and Calibration System before each measurement. The measurement interval was 10 min and the number of data points obtained was 49 at each sampling site. Thermometer (TR-52i; T&D Co.Ltd., Nagano, Japan, the accuracy of $\pm 0.3$ °C at $-20$–80 °C) was used to collect data on the ambient temperature in the range of $-60$–155 °C. The measurement interval was 1 min. TR-73U (T&D Co.,Ltd., Japan) was used to collect data on relative humidity and atmospheric pressure in the range of 10–95%RH and 750–1100hPa, respectively. The measurement interval was 1 min and the number of data points obtained was 540 at each sampling site. An anemometer was used to collect the wind direction and speed data (Kestrel 4500; NIELSEN KELLERMAN Co. Ltd., German, the accuracy of $\pm 5°$ and $\pm 3\%$, respectively) in the range of 0–360° and 0.4–40 m, respectively. The measurement interval was 2 min and the number of data points obtained was 735 at Sanepa. The data at Koteshwor is missing.

*2.2. Analysis*

After sampling, all samples were brought back to the Yokohama environmental science research institute for analysis. BPE-DNPH cartridges and passive samplers were kept cool at approximately 10 °C while traveling.

2.2.1. VOCs and Aldehydes

A VOC analysis system included two parts: a gas chromatography (GC)/a flame ionization detector (FID) (Shimadzu Co., Ltd., Kyoto, Japan) coupled with a concentrator (GL Sciences Inc., Tokyo, Japan) for the C2–C4 components and a GC/mass spectrometer (MS) (Shimadzu Co., Ltd., Kyoto, Japan) coupled with a CC2110 concentrator (GL Sciences Inc., Tokyo, Japan) for remaining components. A 400 mL aliquot of the air sample from each canister was concentrated in each concentrator. The C2–C4 components were rapidly heated to 200°C, transferred to a GC column (TC-BOND Alumina/$Na_2SO_4$, 50 m $\times$ 0.53 mm) with the following GC temperature program: maintained at 50 °C for 6 min; heated from 50 to 120 °C at a rate of 3.0 °C min$^{-1}$; heated from 120 to 150 °C at a rate of 30°C min$^{-1}$; maintained at 150 °C for 5.0 min. Further, quantification was performed using FID. The C2–C4 components detected by FID were ethane, ethylene, propane, acetylene, propylene, isobutene, isobutene, 1-butene, 1,3-butadiene, and butane. The other components were rapidly heated from 60 to 120 °C at a rate of 6°C min$^{-1}$; heated from 120 to 200 °C at a rate of 16 °C min$^{-1}$; maintained at 200 °C for 12 min. Further, quantification was performed using MS. The carrier gas was pure helium (purity > 99.9999%). Each target compound was identified by the ratio of the identified ion to the check ion, its retention time, and the standard gases. Four types of mixed standard gases were used. Ten-point calibration curves including concentrations of 0, 0.02, 0.05, 0.1, 0.2, 0.5, 1, 2, 5, and 10 ppbv were generated, yielding $R^2$ values between 0.995 and 1.00 for the measured compounds. A gas standard (0.5 ppbv) was measured every day to confirm the stability of the system. The detection limits for various compounds were determined based on the gas standard (0.02 ppbv). To calculate the detection limits, the GC/MS and GC/FID analyses were repeated seven times.

The DNPH derivatives for the aldehydes were eluted from the adsorbents by passing the solution through 25% dimethyl sulfoxide in an acetonitrile solution containing 0.085% (*v/v*) phosphoric acid via a syringe and analyzed using a high performance liquid chromatography (1260 Infinity; Agilent Technologies, Co.,Ltd., Santa Clara, CA, USA) equipped with a column (ZORBAX Eclipse Plus C18; 4.6 250 mm) (Agilent Technolo-

gies, Co.,Ltd., Santa Clara, CA, USA). The gradient program was operated first at an acetonitrile/water (55/45 ($v/v$)) for 4 min, second at a linear gradient of acetonitrile/water (90/10 ($v/v$)) for 9 min, and finally at acetonitrile/water (90/10 ($v/v$)) for 5 min. The flow rate was 1 mL·min$^{-1}$. The derivatives were detected at a single wavelength of 365 nm. The limited detections were calculated as being three times the standard deviation obtained from the data of eight replicate measurements.

### 2.2.2. $NO_2$ and $NO_x$

The $NO_2$ and $NO_x$ filter was immersed in 8 mL of ultrapure water for 10 min at room temperature, followed by shaking. The solution was kept cool in the refrigerator after the filter was removed from it. In total, 2 mL of color reagent (sulfanil NEDA solution) was mixed into the cool solution while it was still cool. After 30 min in the refrigerator, the solution was warmed to room temperature and analyzed using a spectrophotometer. A six-point calibration curve including concentrations of 0.1, 0.2, 0.4, 0.6, 0.8, and 1.0 mg·L$^{-1}$ were generated using nitrite ion standard solution (FUJIFILM Wako Pure Chemical Co. Ltd., Osaka, Japan), yielding an $R^2$ value of 0.999. The detection limit (0.0060 mg·L$^{-1}$) was determined based on measuring 10 times repeatedly of the standard solution (0.1 mg·L$^{-1}$).

### 2.2.3. $SO_2$

The $SO_2$ filter was immersed in 10 mL of ultrapure water for 10 min at room temperature, followed by shaking. The filter was removed, and the solution was shaken after 1.0 mL of 1.0% $H_2O_2$ was added. After 10 min, 2 cm of platinum wire (diameter 0.1 mm) was added to the solution. The solution was immersed in water at 50°C for 10 min, then cooled to room temperature, and analyzed using ion chromatography (ICS-1600, DIONEX, Sunnyvale, CA, USA). A six-point calibration curve including concentrations of 0.1, 0.5, 2.5, 10, 20, and 50 mg·L$^{-1}$ were generated using sulfate ion standard solution (FUJIFILM Wako Pure Chemical Co. Ltd., Osaka, Japan), yielding an $R^2$ value of 1.00. The detection limit (0.014 mg·L$^{-1}$) was determined based on measuring 10 times repeatedly the standard solution (0.1 mg·L$^{-1}$). The standard solution (2.5 mg·L$^{-1}$) was measured to confirm the reproducibility.

### 2.2.4. $NH_3$

The $NH_3$ filter was immersed in 10 mL of ultrapure water for 10 min at room temperature, followed by shaking. The filter was removed, and the solution was analyzed using ion chromatography (ICS-1600, DIONEX, Sunnyvale, CA, USA). A five-point calibration curve including concentrations of 0.1, 0.5, 2.5, 5, and 10 mg·L$^{-1}$ were generated using ammonium ion standard solution (FUJIFILM Wako Pure Chemical Co. Ltd., Osaka, Japan), yielding an $R^2$ value of 1.00. The detection limit (0.0038 mg·L$^{-1}$) was determined based on measuring 10 times repeatedly of the standard solution (0.1 mg·L$^{-1}$). The standard solution (2.5 mg·L$^{-1}$) was measured to confirm the reproducibility.

### 2.3. Contribution Analysis in Three Types of Vehicles

The main VOC species in GV exhaust are short-chain hydrocarbons such as isopentane and propane, whereas those in DV exhaust are long-chain hydrocarbons such as dodecane, n-undecane, and n-decane [15]. Meanwhile, acetone, isobutane, isopentane, n-butane, and propane are the VOCs found in LV exhaust [15]. Sakamoto et al. [16] proposed that the original fuel composition and background level can be used to estimate the mixing ratios of compressed natural gas, LPG, and gasoline. Taking advantage of this idea, the contributions of different types of vehicles were calculated using the original tailpipe composition [15] and the following equation:

$$\begin{pmatrix} B_{Koteshwor} \\ P_{Koteshwor} \\ N_{Koteshwor} \end{pmatrix} = \alpha \begin{pmatrix} B_{LV} \\ P_{LV} \\ N_{LV} \end{pmatrix} + \beta \begin{pmatrix} B_{GV} \\ P_{GV} \\ N_{GV} \end{pmatrix} + \gamma \begin{pmatrix} B_{DV} \\ P_{DV} \\ N_{DV} \end{pmatrix} \tag{1}$$

where B, P, and N are the concentrations of isobutane, isopentane, and n-nonane, respectively, and the coefficients α, β, and γ are relative impacts on LV, GV, and DV, respectively. The coefficients α, β, and γ are obtained by solving the simultaneous Equation (1). The contribution rate of LV, GV, and LV was calculated by dividing α, β, and γ by their sum, respectively. The background levels were not considered.

### 2.4. Estimation of the Photochemical Ozone Production

The photochemical reactivity of VOC species, as well as the level of concentration, has a significant influence on ozone production. OFP and OH radical loss rate ($L^{OH}$) have been used to describe the contribution of VOC species to ozone production [12,13]. The OFP value represents the maximum amount of ozone that can be generated from VOCs, whereas the $L^{OH}$ value represents how fast the OH radical reacts with VOCs. OFP is likely to overestimate the VOC contribution to ozone production, whereas OH radical loss rate is likely to underestimate its contribution. In addition, the MIR values [17] used for the calculation of OFP were calculated under the $NO_x$ concentration at which the ozone concentration increased most significantly. OH radical loss rate is an indicator to accurately represent ozone production for 1–3 h. However, because of the stagnant air in the Kathmandu Valley, air pollutants have a longer transport time. According to Sharma et al. [18], the actual time taken by the urban air mass to reach the rural was ~4 days during the monsoon season. Winter is expected to last longer than monsoon. As a result, OFP was chosen as an indicator in this study to accurately represent ozone production. OFP is calculated by multiplying VOC concentration by the MIR value as follows:

$$OFP_i = [VOC]_i \times MIR_i \qquad (2)$$

### 3. Results and Discussion

### 3.1. Meteorological Characteristics

Figure 1 shows variations of temperature, humidity, and atmospheric pressure on each sampling day at Koteshwor and Sanepa. On the sampling day (10 March) at Sanepa, increasing temperature caused thermal convection to develop and decline of humidity and air pressure in the morning. Subsequently, a valley wind started to blow around noon, when temperature decreased whereas humidity increased. Wind data in Sanepa supported this. Figure 2 shows variations of wind direction (WD) and wind speed (WS) from 9–10 March in Sanepa. WS rapidly became strong and WD changed from southerly to northerly around noon. On the contrary, on the sampling day (7 March) at Koteshwor, thermal convection developed until approximately 13:00, and wind was estimated to have become calm at around 15:00. It was anticipated that a valley wind was weak because the temperature did not decrease around noon (wind data at Koteshwor was missing).

### 3.2. Characteristics of Air Pollutants

Figure 3 shows the time series of VOC concentrations and their ratio at Koteshwor and Sanepa. It is noted that the sampling day at Koteshwor was different from that at Sanepa. The VOC concentration was much higher at Koteshwor than Sanepa. Alkanes were the most abundant at Koteshwor (46–102 ppbv, 28–34%) and Sanepa (12–37 ppbv, 29–32%), followed by alkenes (36–67 ppbv, 12–19%) and aromatics (29–71 ppbv, 18–22%) at Koteshwor, and aldehydes (5.1–22 ppbv, 12–21%) and alkenes (4.7–22 ppbv, 12–19%) in Sanepa. Maximum, minimum, and mean concentrations with standard deviation are displayed in Table 2. On average, ethylene (29 ppbv), formaldehyde (23 ppbv), and acetylene (21 ppbv) were abundant at Koteshwor, accounting for ~30% of total VOC. At 9:00 a.m., ethylene (32 ppbv), formaldehyde (24 ppbv), and acetylene (21 ppbv) were abundant at Koteshwor, accounting for 28% of total VOC, while formaldehyde (12 ppbv), ethylene (12 ppbv), and acetylene (7.7 ppbv) were abundant at Sanepa, accounting for 28% of total VOC. It was estimated that these species were widespread across the Kathmandu Valley. It was discovered that vehicle exhaust emissions have a significant impact on the early morning ambient air. The

total VOC concentration in Koteshwor was higher at 11:30 than it was at 9:00. As the top 10 VOC species with high increase concentrations were isopentane, toluene, ethylene, and acetylene, which are abundant in GV exhaust, it was estimated that GV exhaust increased at 11:30 and exceeded the decrease due to the mixing layer. After 11:30 a.m., oxygenated VOCs such as acetone, formaldehyde, and acetaldehyde were abundant. It was assumed that these species were the secondary products of photochemical reactions. Ethane, which is not found in gasoline, was also abundant. It was speculated that ethane was emitted from domestic sources because LPG was widely used as fuel.

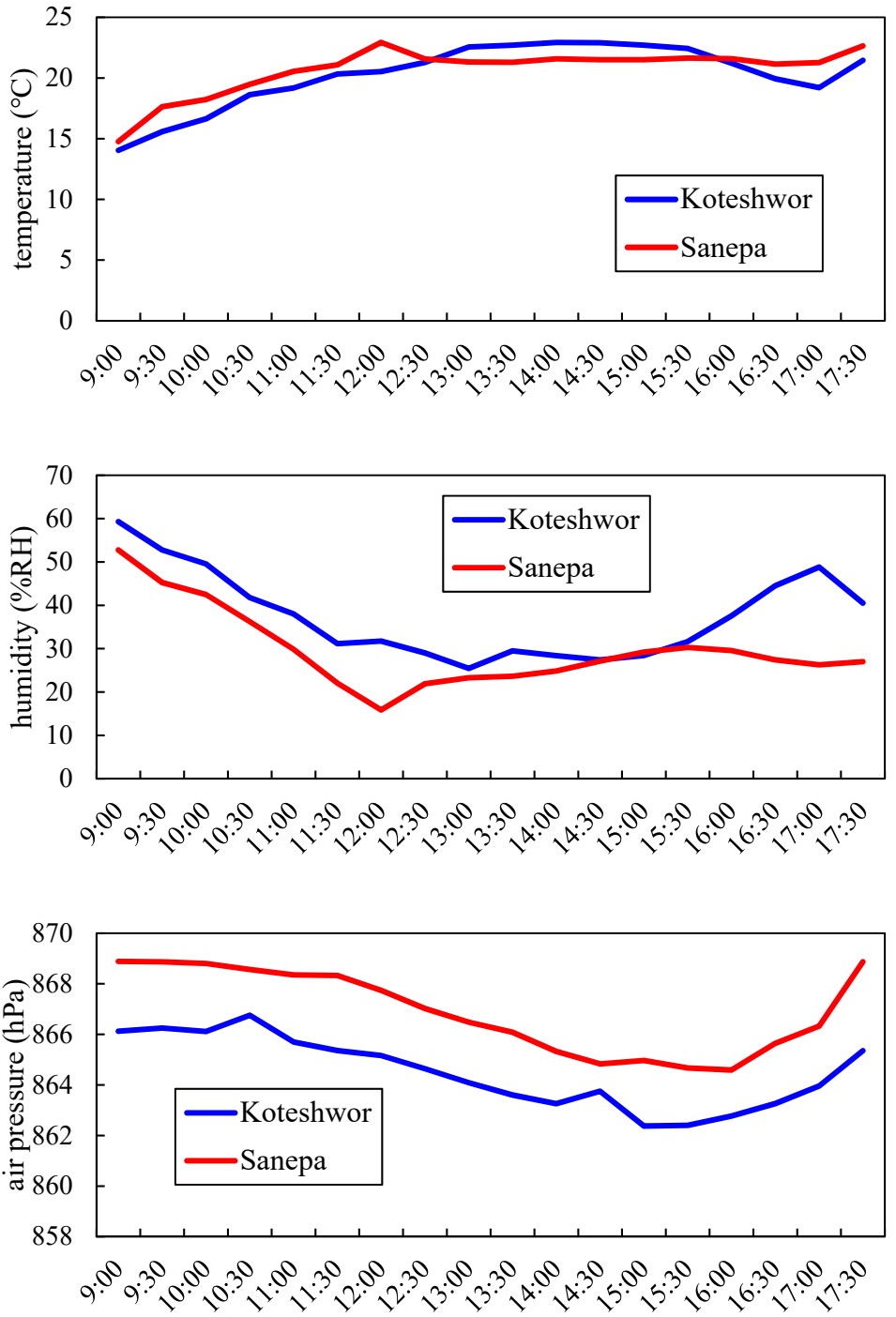

**Figure 1.** Variation of temperature, humidity, and atmospheric pressure at each sampling day.

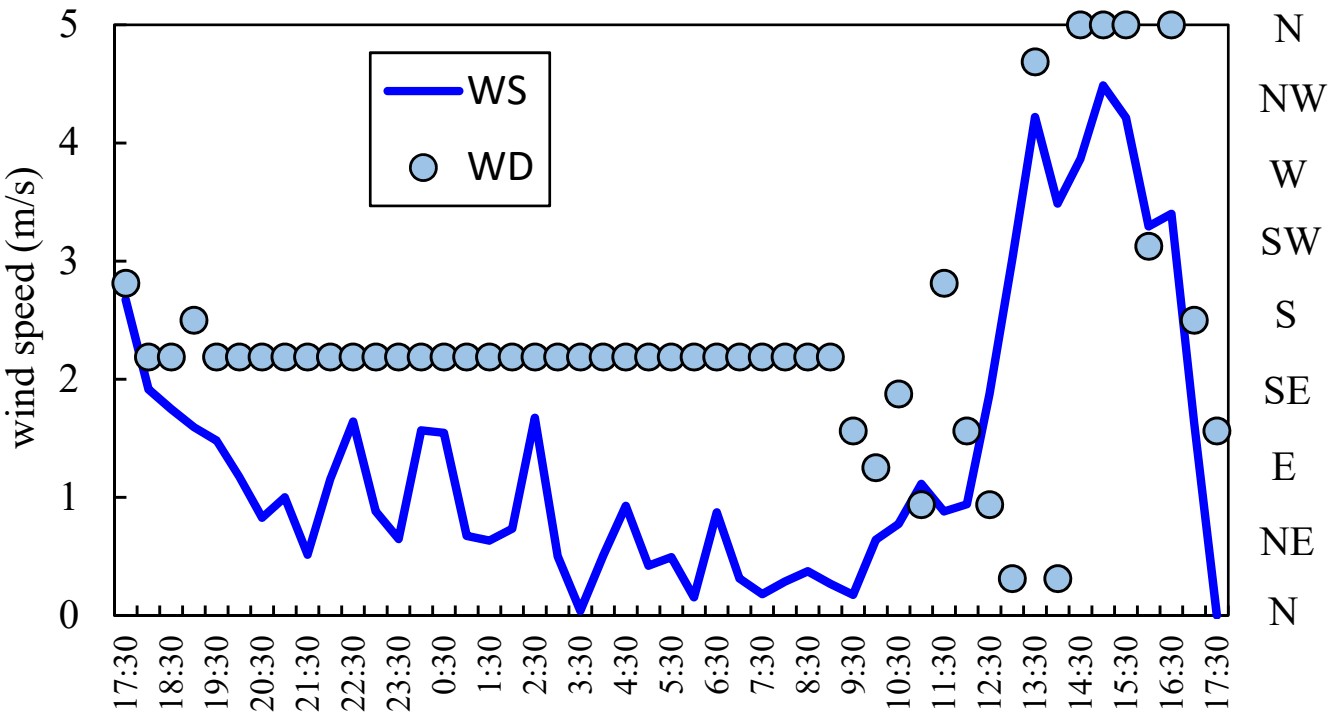

**Figure 2.** Variation of wind direction and wind speed at sampling day in Sanepa.

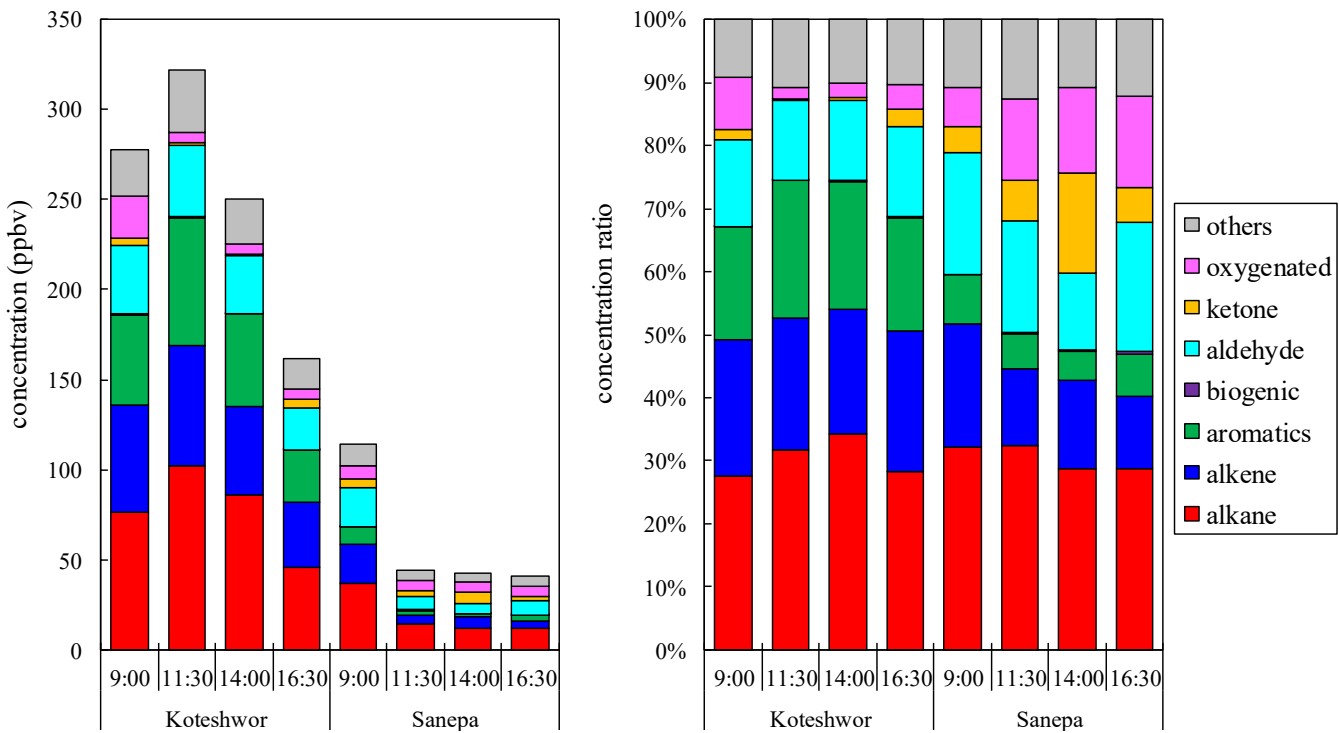

**Figure 3.** The time series of the VOCs concentration (left) and its ratio (right).

**Table 2.** Maximum, minimum, and mean concentrations (ppbv) of VOCs at Koteshwor and Sanepa.

| | Koteshwor | | | Sanepa | | | SD [1] |
| --- | --- | --- | --- | --- | --- | --- | --- |
| | Maximum | Minimum | Mean | Maximum | Minimum | Mean | (*n* = 5) |
| Ethane | 8.3 | 6.3 | 7.2 | 6.7 | 4.3 | 5.0 | 0.26 |
| Propane | 7.3 | 2.1 | 5.3 | 7.2 | 2.1 | 3.7 | 0.27 |
| n-butane | 7.6 | 2.8 | 6.0 | 5.9 | 1.3 | 2.6 | 0.21 |
| Isobutane | 5.9 | 1.3 | 3.6 | 5.6 | 0.51 | 1.9 | 0.093 |
| n-pentane | 7.0 | 2.8 | 4.8 | 1.1 | 0.27 | 0.55 | 0.036 |
| Isopentane | 27 | 9.8 | 18 | 4.1 | 0.91 | 1.9 | 0.015 |
| Cyclohexane | 4.0 | 1.8 | 2.9 | 0.51 | 0.16 | 0.25 | 0.038 |
| n-nonane | 1.7 | 0.28 | 0.74 | 0.099 | 0.046 | 0.059 | 0.030 |
| n-decane | 1.7 | 0.044 | 0.53 | N.D. [2] | N.D. | N.D. | 0.029 |
| n-undecane | 1.3 | N.D. | 0.42 | 0.05 | N.D. | N.D. | 0.033 |
| Ethylene | 38 | 21 | 29 | 12 | 2.3 | 5.0 | 0.39 |
| 1-butene | 2.2 | 1.1 | 1.5 | 1.1 | 0.09 | 0.39 | 0.060 |
| Acetylene | 30 | 13 | 21 | 7.7 | 0.49 | 2.6 | 0.33 |
| Acetone | 3.9 | 0.45 | 2.1 | 5.8 | 1.9 | 3.7 | 0.30 |
| Benzene | 7.9 | 3.7 | 5.6 | 2.1 | 0.30 | 0.79 | 0.047 |
| Toluene | 25 | 11 | 17 | 3.1 | 0.62 | 1.3 | 0.059 |
| 1,2,4-trimethylbenzene | 4.0 | 0.90 | 2.8 | 0.29 | 0.054 | 0.11 | 0.036 |
| 1,2,3-trimethylbenzene | 1.0 | 0.24 | 0.72 | 0.084 | 0.042 | 0.052 | 0.028 |
| Formaldehyde | 27 | 17 | 23 | 12 | 3.7 | 6.5 | 0.00032 |
| Acetaldehyde | 14 | 6.3 | 11 | 10 | 1.5 | 4.5 | 0.00012 |

[1] SD were calculated for the samples from Koteshwor and Sanepa combined. [2] N.D.: no detection.

Figure 4 shows the variation in air pollution levels of $PM_{2.5}$, CO, $O_3$, $NH_3$, NO, and $NO_2$ at Koteshwor and Sanepa. At both sites, the level of $PM_{2.5}$ reached the peak under the influence of heavy traffic and the developing inversion layer at 9:10, with 237 and 150 $\mu g \cdot m^{-3}$ at Koteshwor and Sanepa, respectively. The level of $PM_{2.5}$ gradually decreased after 9:00 as the mixing layer developed. At Koteshwor and Sanepa, the mean concentration of $PM_{2.5}$ during the day was 107 ± 36 and 59 ± 46 $\mu g\ m^{-3}$, respectively. Koteshwor had much higher CO levels than Sanepa. Though the fluctuation amplitude of CO was larger than $PM_{2.5}$, the variation in CO concentrations was consistent with that of $PM_{2.5}$. Sanepa had a higher level of $O_3$ during the day, with an average value of ~60 ppbv, than Koteshwor, which had a value of ~10 ppbv. The WS at Sanepa was 0.7, 2.7, and 3.9 $m \cdot s^{-1}$ at 9:00, 11:30, and 14:00, respectively. The WS became strong in the afternoon and peaked at 14:00. However, there was no significant variation in $O_3$ concentration based on WS. At Koteshwor, the level of NO was high, whereas the level of $O_3$ was low. It was deemed that $O_3$ concentration did not increase because of the loss from the reaction of $O_3$ with NO at Koteshwor, whereas $O_3$ generation was promoted by the photochemical reaction because of lower $NO_x$ levels at Sanepa.

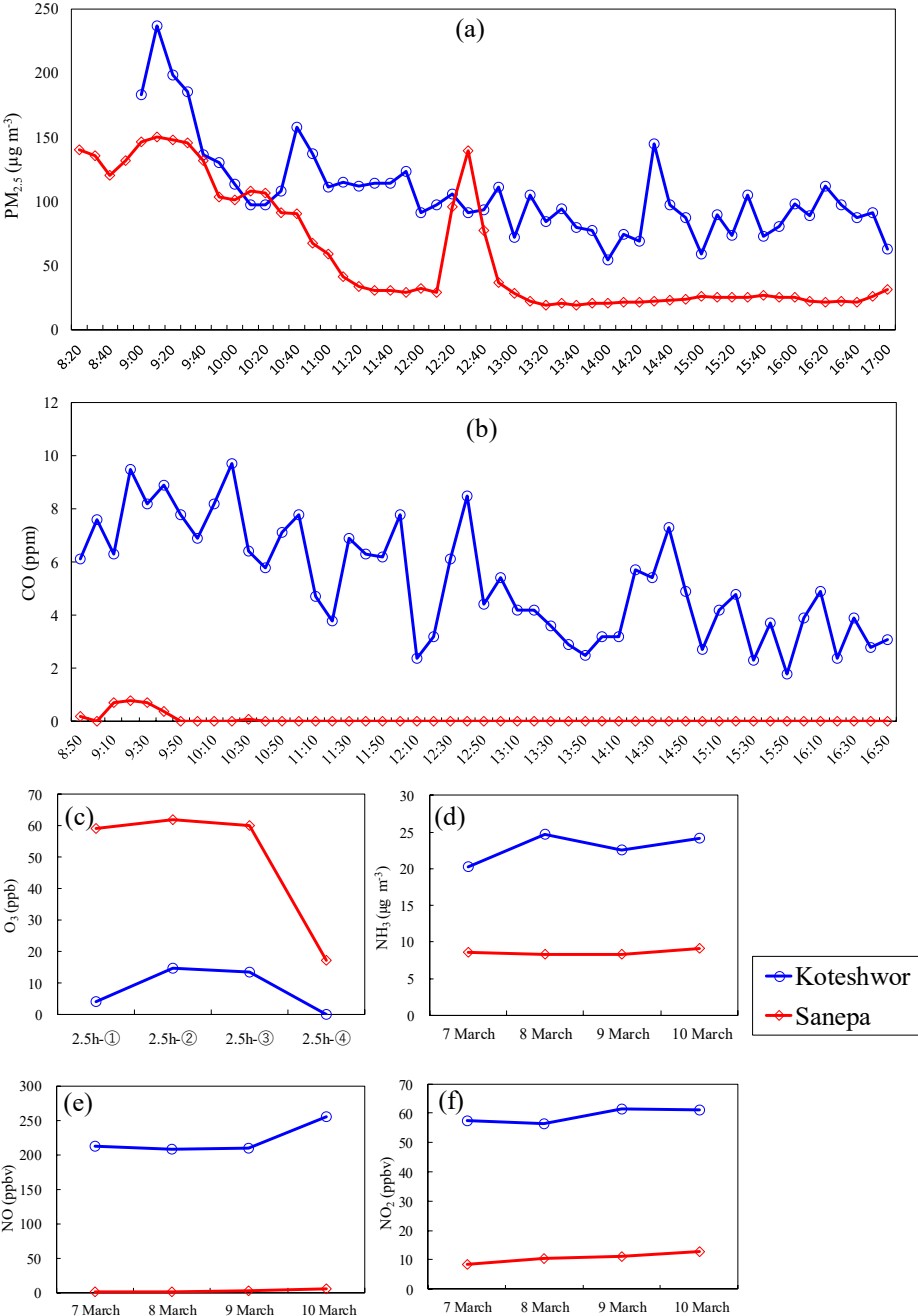

**Figure 4.** Variation of the levels of air pollutants: (**a**) $PM_{2.5}$, (**b**) CO, (**c**) $O_3$, (**d**) $NH_3$, (**e**) NO, and (**f**) $NO_2$.

Throughout the investigation days, the $NH_3$ concentrations at Koteshwor and Sanepa were kept at ~30 and 11 ppbv, respectively. Ammonium salt is formed by reacting $NH_3$ with sulfuric and nitric acids. Ammonium salt is a particle, whereas $NH_3$ is a gas. The concentration levels of gaseous ammonia and ammonium salt were reported to be the same by Sakurai et al. [19]. Granulation is promoted as $NH_3$ concentration increases because this granulation reaction is reversible. Kathmandu is assumed to be in the atmospheric environment where the particle formation reaction between $NH_3$ and nitric acid is likely to occur. The $SO_2$ level was below the methodological detection limit.

### 3.3. Air Pollutants Related to the Vehicular Exhaust Emissions

The sampling site at Koteshwor is located between busy traffic roads. Therefore, it was assumed that air pollutants in Koteshwor consisted of only three types of vehicles (GV, DV, and LV) exhaust. Though a certain number of pollutants are expected to reach Kathmandu from outside Kathmandu through long-distance transport/advection processes, it was neglected in this study because the contribution of the background was below 30% in winter [20].

Figure 5 shows the contributions in three types of vehicles in Koteshwor. Table 3 shows the values of coefficients calculated by Equation (1). The average contributions of GV, LV, and DV exhaust were 68%, 22%, and 10%, respectively. At 11:30 and 16:30, GV exhaust contributed more than 85% of the total. The contribution of LV exhaust, however, was greatest at 9:00. It was probably affected by the LPG tank which stayed for a few minutes within the sampling duration near the sampling site due to the traffic congestion of commuting hours.

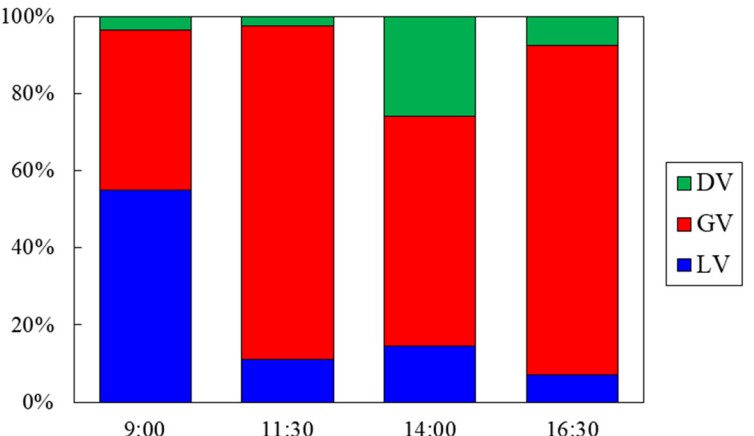

**Figure 5.** Contribution in three types of vehicles (LV, GV, and DV) at Koteshwor.

**Table 3.** Concentrations (ppbv) of isobutane, isopentane, and n-nonane, and values of $\alpha$ (LV), $\beta$ (DV) and $\gamma$ (DV) calculated by Equation (1).

|  | Isobutane | Isopentane | n-Nonane | $\alpha$ | $\beta$ | $\gamma$ |
|---|---|---|---|---|---|---|
| 9:00 | 5.9 | 15 | 0.45 | 0.0114 | 0.00867 | 0.000732 |
| 11:30 | 3.7 | 27 | 0.47 | 0.00227 | 0.0177 | 0.000512 |
| 14:00 | 3.5 | 20 | 1.7 | 0.00314 | 0.0129 | 0.00560 |
| 16:30 | 1.3 | 9.8 | 0.28 | 0.000535 | 0.00660 | 0.000578 |
|  |  |  | Tailpipe [15] |  |  |  |
| LV | 392.6 | 177.7 | 9.3 |  |  |  |
| GV | 158.1 | 1475.1 | 17.6 |  |  |  |
| DV | 37.5 | 23.1 | 266.3 |  |  |  |

Table 4 shows the correlation coefficients among the abundant VOC species in LV, GV, and DV exhaust, ethylene, acetylene, $PM_{2.5}$, and CO in Koteshwor. It was noted that this analysis had limitations because the sample size ($n = 4$) was too small for the correlation analysis. Therefore, the results of this analysis were compared with the known tailpipe compositions and contribution rates in three types of vehicles. Ethylene and acetylene are common internal combustion engine products in the urban atmosphere. Isopentane, which is most abundant in GV exhaust, correlated well with pentane, cyclohexane, and benzene, which are the top 10 VOC species in GV exhaust [15]. Propane had a moderate correlation with isopentane, suggesting that propane was also emitted from DV and LV. Actually, propane also correlated with VOC species in DV exhaust such as 1,2,4-trimethylbenzene and 1,2,3-trimethylbenzene and those in LV exhaust such as isobutane and n-butane.

Decane, which is abundant in DV exhaust, correlated well with n-nonane. This indicated that those species were emitted by DV. The components correlated with ethylene and acetylene, which are combustion tracers, corresponded to those correlated with isopentane. CO also correlated well with ethylene, acetylene, benzene, and toluene, indicating that GV exhaust has the greatest impact. These findings indicate that pentane, isopentane, and benzene were emitted by vehicular exhaust emissions rather than gasoline evaporation. Despite the absence of samples, the correlation coefficients agreed with the results of the original exhaust components and fuel contributions. Meanwhile, $PM_{2.5}$ was linked to 1-butene and methylethylketone, both of which are abundant in DV and GV exhaust, respectively. 1-butene did not correlate with n-nonane and n-decane, but with propane. Additionally, methylethylketone conversely had a negative correlation coefficient with each VOC, as well as acetone. Though this might be affected by photochemical reaction, VOC data set was too small ($n = 4$) to evaluate. The estimation requires further investigation with higher temporal resolution.

**Table 4.** The correlation coefficients between air pollutants in Koteshwor ($n = 4$).

| | Ethylene | Acetylene | Isopentane | n-Decane | $PM_{2.5}$ | CO |
|---|---|---|---|---|---|---|
| Acetylene | 0.94 | | | | | |
| Isopentane | 0.82 | 0.96 | | | | |
| n-decane | −0.20 | 0.03 | 0.24 | | | |
| $PM_{2.5}$ | 0.32 | −0.01 | −0.27 | −0.55 | | |
| CO | 0.95 | 0.80 | 0.61 | −0.45 | 0.56 | |
| Propane | 0.73 | 0.67 | 0.60 | 0.31 | 0.40 | 0.65 |
| Isobutane | 0.65 | 0.47 | 0.32 | 0.06 | 0.70 | 0.67 |
| n-butane | 0.83 | 0.78 | 0.68 | 0.21 | 0.40 | 0.75 |
| Pentane | 0.88 | 0.99 | 0.99 | 0.15 | −0.15 | 0.70 |
| Cyclohexane | 0.83 | 0.97 | 1.0 | 0.25 | −0.23 | 0.62 |
| n-nonane | −0.16 | 0.08 | 0.29 | 1.0 | −0.55 | −0.41 |
| 1-butene | 0.43 | 0.17 | −0.03 | −0.12 | 0.88 | 0.55 |
| Benzene | 0.99 | 0.97 | 0.88 | −0.17 | 0.20 | 0.91 |
| Toluene | 0.95 | 1.00 | 0.95 | −0.03 | 0.02 | 0.82 |
| 1,2,4-trimethylbenzene | 0.78 | 0.89 | 0.92 | 0.45 | −0.11 | 0.56 |
| 1,2,3-trimethylbenzene | 0.76 | 0.88 | 0.92 | 0.48 | −0.13 | 0.54 |
| m-ethyltoluene | 0.82 | 0.93 | 0.95 | 0.39 | −0.12 | 0.61 |
| m-diethylbenzene | 0.49 | 0.75 | 0.90 | 0.53 | −0.63 | 0.21 |
| p-diethylbenzene | 0.55 | 0.74 | 0.85 | 0.69 | −0.33 | 0.29 |
| Acetone | −0.47 | −0.71 | −0.86 | −0.70 | 0.53 | -0.18 |
| Methylethylketone | −0.06 | -0.38 | −0.60 | −0.47 | 0.93 | 0.21 |

### 3.4. Ratio of Ethylene to Acetylene

The ethylene to acetylene (E/A) ratio is used to determine whether a vehicle is equipped with a well-maintained catalyst. Its ratio of well-maintained catalyst-equipped vehicles is three or higher [21] whereas that of non-catalyst cars is closer to one [22]. The E/A ratios at Koteshwor were 1.5, 1.3, 1.3, and 1.6 at 9:00, 11:30, 14:00, and 16:30, respectively. In Kathmandu, it was revealed that most vehicles were not equipped with well-maintained catalysts.

### 3.5. Comparison to the Previous Studies

Table 5 compares the VOC concentrations at Koteshwor with those in previous studies [14,18]. Regardless of the increase in the number of vehicles, the levels of combustion and evaporation components were reduced compared with the results in 1998. It was assumed that the level of air pollutants in vehicular exhaust emissions decreased due to the measures implemented after 1998. Table 6 shows the ratios of burned components to unburned components. In this study, the ratios of gasoline burned components such as ethylene and acetylene to gasoline unburned components such as isobutane, n-butane, isopentane, and n-pentane, except isopentane at 11:30, were greater than in 1998. It indicated improved engine combustion efficiencies. Furthermore, these findings supported the discussion in Section 3.3 that VOC species from vehicles were emitted by vehicular exhaust emissions rather than fuel evaporation. On the contrary, the ratios of burned components to unburned components such as ethane and propane, which is abundant in LPG, became smaller, implying that LPG was more widely used. Moreover, the concentration increase was only found in cyclohexane, which is a component of DV exhaust, implying that the impact of DV exhaust has increased. When the results from 2003 were compared, the levels of all components, except hexane, increased, especially isopentane, which is abundant in GV exhaust. It was considered that an increase in the number of GV offset the effect of the measures and exceeded it, increasing the level of air pollutants. Meanwhile, the increase in the levels of 2-methylpentane and 3-methylpentane, which are components in DV exhaust, was likely to be caused by the increase in the number of DV.

**Table 5.** Comparison of the VOC concentrations (ppbv) in this study with those in the previous studies.

| | Weekdays AverAge at Putalisadak (1998) [18] | Mean at Eight RoadSide (2003) [14] | Koteshwor 9:00 | Koteshwor 11:30 |
|---|---|---|---|---|
| Ethane | 7.86 | | 8.35 | 7.58 |
| Ethylene | 59.68 | | 31.91 | 38.36 |
| Acetylene | 43.51 | | 20.85 | 29.98 |
| Propane | 7.01 | 2.42 | 7.31 | 5.85 |
| Propylene | 15.23 | 6.97 | 11.93 | 9.98 |
| Isobutene | 21.02 | 2.17 | 5.87 | 3.71 |
| n-butane | 53.87 | 2.86 | 7.62 | 7.24 |
| trans-2-butene | 0.76 | | 0.85 | 0.70 |
| 1-butene | 2.94 | | 2.19 | 1.34 |
| cis-2-butene | 0.52 | | 0.72 | 0.51 |
| 1,3-butadiene | 4.18 | | 1.55 | 1.48 |
| Isopentane | 33.48 | 8.07 | 14.83 | 26.53 |
| Pentane | 32.11 | 4.08 | 4.26 | 6.95 |
| Cyclopentane | 5.65 | | 0.67 | 1.07 |
| Cyclopentene | 0.46 | | | |
| 3-methyl-1-butene | 0.51 | | 0.28 | 0.40 |
| trans-2-pentene | 0.8 | | 1.12 | 2.00 |
| 2-methyl-2-butene | 0.67 | | 2.04 | 3.65 |
| 1-pentene | 1.75 | | 0.48 | 0.61 |
| cis-2-pentene | 0.9 | | 0.56 | 0.93 |
| Isoprene | 0.36 | | 0.44 | 0.33 |

**Table 5.** *Cont.*

| | Weekdays AverAge at Putalisadak (1998) [18] | Mean at Eight RoadSide (2003) [14] | Koteshwor 9:00 | Koteshwor 11:30 |
|---|---|---|---|---|
| Hexane | 18.09 | 2.67 | 1.43 | 2.23 |
| Methylcyclopentane | 23.84 | | 1.97 | 3.11 |
| 3-methylpentane | 17.69 | 2.16 | 2.62 | 4.20 |
| 2,2-dimethylbutane | 3.92 | | 1.74 | 2.90 |
| 2-methylpentane | 21.15 | 3.09 | 5.01 | 8.21 |
| 2,3-dimethylbutane | 12.59 | | 1.78 | 2.84 |
| Cyclohexane | 1.85 | | 2.61 | 4.01 |

**Table 6.** Ratios of burned components to unburned components.

| | | Weekdays Average at Putalisadak (1998) [18] | Koteshwor 9:00 | Koteshwor 11:30 |
|---|---|---|---|---|
| Ethylene/ | Ethane | 7.59 | 3.82 | 5.06 |
| | Propane | 8.51 | 4.37 | 6.56 |
| | Isobutene | 2.84 | 5.43 | 10.4 |
| | n-butane | 1.11 | 4.19 | 5.30 |
| | Isopentane | 1.78 | 2.15 | 1.45 |
| | n-pentane | 1.86 | 7.49 | 5.52 |
| Acetylene/ | Ethane | 5.54 | 2.50 | 3.95 |
| | Propane | 6.21 | 2.85 | 5.13 |
| | Isobutane | 2.07 | 3.55 | 8.09 |
| | n-butane | 0.81 | 2.74 | 4.14 |
| | Isopentane | 1.30 | 1.41 | 1.13 |
| | n-pentane | 1.36 | 4.89 | 4.31 |

*3.6. The VOCs Impact on Photochemical Ozone Production*

Figure 6 shows the time series of OFP. Aromatics and alkenes accounted for 66–79% and 43–59% of total OFP in the two sampling sites, respectively. Aromatics and alkenes were found to contribute significantly to ozone production. When comparing OFP in Koteshwor to that in Sanepa, aromatics contributed more in Koteshwor than in Sanepa, while aldehyde contributed more in Sanepa than in Koteshwor. Aldehyde accounted for 15–19% and 30–51% of the total in Koteshwor and Sanepa, respectively. The photochemical reaction in Sanepa was expected to result in a decrease in aromatics and an increase in aldehyde. In terms of individual components, ethylene, propylene, toluene, and m-xylene all made significant contributions to photochemical ozone production. Those four components accounted for 39–47% and 33–49% in Koteshwor and Sanepa, respectively. Since ethylene, acetylene, and toluene correlated well with isopentane which is abundant in GV exhaust, it was estimated that the main source of those components was GV exhaust.

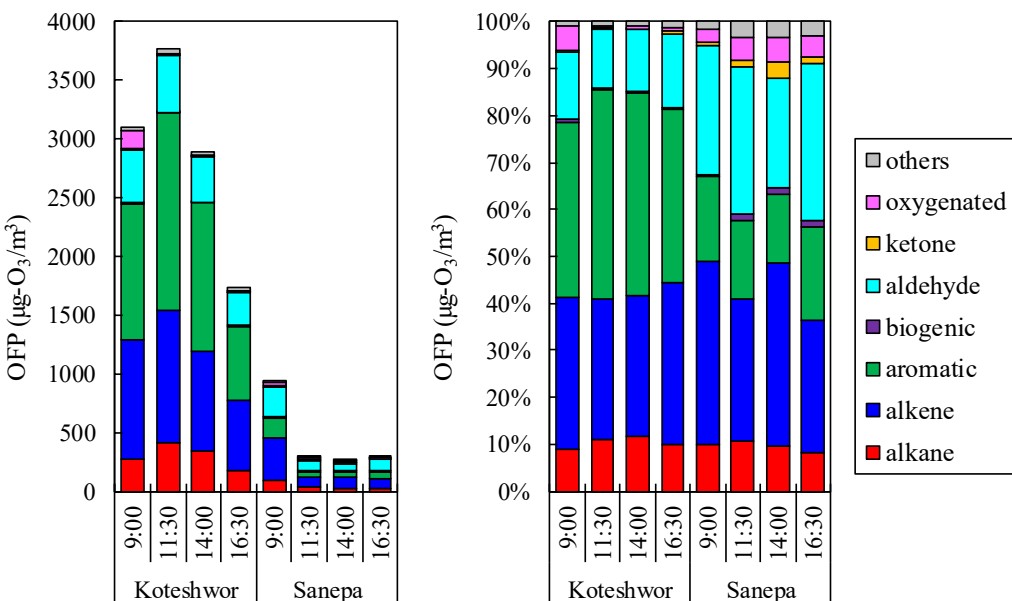

**Figure 6.** Time series of ozone formation potential at Koteshwor and Sanepa.

## 4. Conclusions

We investigated the impact of vehicular-related emissions on air pollution, as well as the contribution of VOCs to photochemical ozone production in the Kathmandu Valley. Koteshwor and Sanepa had the highest contributions of alkanes. The level of $PM_{2.5}$ reached a peak at 9:10 under heavy traffic influence and the developing inversion layer. The CO concentration time series was also similar to the $PM_{2.5}$ mass concentration time series. The levels of $O_3$ and $NO_2$ were higher at Sanepa than Koteshwor, whereas the level of NO was much higher at Koteshwor than Sanepa, implying that $O_3$ loss was due to the reaction with NO at Koteshwor. Additionally, the Kathmandu Valley was estimated to be in the atmospheric environment where the particle formation reaction between $NH_3$, sulfuric, and nitric acid may occur because the level of $NH_3$ was high.

In the Kathmandu Valley, it was estimated that GV exhaust was attributed to gaseous air pollutants such as VOCs significantly, based on the analysis combined correlation analysis with contributions in three types of vehicles. However, this analysis has limitations because the sample size was very small, and further investigation is required for source attribution analysis such as PMF. The ratio of ethylene to acetylene showed that most vehicles in Kathmandu were not equipped with a well-maintained catalyst. Compared to previous studies, it was considered that an increase in the number of GV offset the effect of the measures and exceeded it, increasing the level of air pollutants.

For photochemical ozone production, aromatics and alkenes were highest for OFP. Individual components, such as ethylene, propylene, toluene, and m-xylene, were found to contribute significantly to photochemical ozone production. As those components correlated well with isopentane, which is abundant in GV exhaust, it was assumed that GV exhaust was the primary source of those components. From the point of view of photochemical ozone production, strategies for regulating GV exhaust emissions are critical for controlling the photochemical smog in the Kathmandu Valley.

**Supplementary Materials:** The following are available online at https://www.mdpi.com/article/10.3390/atmos12101322/s1, Figure S1: Map of Kathmandu, Nepal, including two sampling sites. Reproduced with permission from Japan International Corporation Agency (JICA); Figure S2: Sampling at Koteshwor site; Figure S3: Sampling at Sanepa site.

**Author Contributions:** S.N. designed the study, reviewed the manuscript, and exercised overall supervisory control. Y.K. analyzed VOCs using GC-FID/MS. Y.I. analyzed $NO_2$, $NO_x$, $SO_2$, $O_3$, and $NH_3$ using IC. M.U. contributed to the contribution analysis in the three types of vehicles.

Y.F. conducted the investigation, analyzed aldehydes using a high performance liquid chromatography, analyzed the data, edited the entire manuscript, and administrated this project. All authors have read and agreed to the published version of the manuscript.

**Funding:** This research was conducted as the project of Japan International Cooperation Agency. This research received no external funding.

**Data Availability Statement:** The data of this atmospheric investigation in the Kathmandu Valley was offered by Japan International Cooperation Agency.

**Acknowledgments:** The author would like to acknowledge Naoki Nishimura, Marina Togo, and Jeevan Shrestha for their assistance in conducting the field campaigns and thank everyone at the Tokyo Metropolitan Research Institute for Environmental Protection for giving me valuable advice.

**Conflicts of Interest:** The authors declare no conflict of interest.

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
