# Peer review of "Investigation of Air Pollutants Related to the Vehicular Exhaust Emissions in the Kathmandu Valley, Nepal"

_atmosphere, doi:10.3390/atmos12101322_

Round 1

Reviewer 1 Report

Introduction

Your references are very old. Update your scientific base, citing new references, such as:

Santana, J.C.C.; Miranda, A.C.; Souza, L.; Yamamura, C.L.K.; Coelho, D.d.F.; Tambourgi, E.B.; Berssaneti, F.T.; Ho, L.L. Clean Production of Biofuel from Waste Cooking Oil to Reduce Emissions, Fuel Cost, and Respiratory Disease Hospitalizations. Sustainability 2021, 13, 9185. https://doi.org/10.3390/su13169185

Introduction and methods

Please quote the environmental regulations (for gaseous emissions) of your country in your text and also use the WHO standards for gaseous emissions. Thus, we are able to find out if emissions in Kathmandu are outside environmental standards.

Results and discussion

These arguments do not guarantee that emissions come solely from burning gasoline. Gasoline only generates low carbon emissions. If there is a mixture, you cannot say that gasoline is the main responsible for emissions

In general, the least efficient are diesel engines and most emissions are derived from their incomplete burning. Your table 3 corroborates this. Since the incomplete burning of gasoline (octane/heptane/hexane) only generate low carbon components. The heavier components (mainly cyclic hydrocarbons) are derived from diesel oils (in addition to the light components).

Reviewer 2 Report

The article investigates important sources of air pollution in Kathmandu, Nepal. I found the article interesting overall, but I think the period used for the analyses is too short, which can provide equivocated conclusions about air pollution sources in the region. It would be better to increase the period of study. Another point is that the time of measurement in the two sites is different, which also does not allow a correct comparison between locations. Other specific comments are presented below.

Page 1, lines 36-37. I wouldn't say "air emissions". Something like “dilution of air pollutants” would sound better.

Page 1, line 41. Change ”…8:00, the level…” to “…8:00, when the level…”

Page 2, lines 55 – 56. Define LPG. Also, it would be better to say “… in domestic activities” or something like that.

Page 2, line 60. You mention some primary sources. What is the secondary source?

Page 2, line 78. I think you meant to say “air quality” instead of “air pollution”.

Page 4, line 127. Why you didn't use the same period in your analysis? Same equipments used in the two sites? Don’t you think comparison will be not fair?  Please, explain.

Page 4, lines 140-141. That is not right. Thermometers measures only Temperature. Please, define the equipments correctly.

Page 4, line 152. The authors mention that “The details are described in Fukusaki et al (2021)”. You should explain it here since it is important for your analysis.

Page 6, lines 194-197. The authors mention that “Because air pollutants are accumulated in the Kathmandu Valley, long-term photochemical ozone production is just as important as short-term photochemical ozone production. Therefore, OFP and OH radical loss rate were evaluated in this study.” I didn't understand the connection. Could you explain it better?

Page 6, line 202. When you say 78ppbv, 31%, Is this an average value using four different hours (9:00, 11:30, 14:00, 16:30)? Please explain in the text. It’s not easy to see in your graphics. The same applies to the rest of your results in the same paragraph.

Page 9, line 247. Define MDL.

Reviewer 3 Report

This manuscript presents an analysis of the sources of a comprehensive list of air pollutants, including speciated VOCs, based on the measurements made at two locations in the Kathmandu valley. Although the study does not present new findings or methods, the topic is still worth publishing because the study is likely to fulfill the important data gap in this region. However, the manuscript suffers from several serious flaws, which make it unsuitable for publishing in its current form. The study methods (sample size, instrumentation, calibration, and other QAQC) are not clearly presented. The manuscript lacks uncertainty analysis. Importantly, many claims made by the authors are not supported by the data. Therefore, I recommend reconsideration of the manuscript following major revision. My specific comments on the manuscript are listed below:

  1. Line 13 (Abstract): What’s the meaning of the term “secondary production”? Do the authors mean “secondary pollutants”? Please correct.
  2. Line 38-40: Please cite the relevant source.
  3. Line 45-47; Line 49-50: Please cite the source of the data.
  4. Line 52: “The Kathmandu Valley has no significant VOC emission sources”. I do not agree with this statement. There are a lot of gasoline vehicles and motorcycles in the valley, which I believe, emit a lot of VOCs. On what basis do the authors claim that there are no significant VOC emission sources in the Kathmandu Valley? Please clarify.
  5. Line 68-71: I suggest the authors move this statement to the relevant place in the “discussion” sections.
  6. Line 76: change to “Nepal Vehicular Mass Emission Standard”.
  7. Line 77-78: “there has been no improvement in the air pollution”; this statement should be supported by data. Please cite relevant source/s that show that the valley's air quality is degrading (or not improving) despite various policy initiatives.
  8. Line 81-82: Please cite the source of the information.
  9. Line 92: It’s better to replace the word “initiate” with “participate in”.
  10. Line 93: I suggest the authors replace the term “secondary production” with “secondary pollutant” here and throughout the manuscript.
  11. Line 94-95: “In the future, the reduction in NOx concentration due to vehicle-related measures may cause serious photochemical smog”. This statement is misleading. The improvement in NOx emission control can be expected mainly with the improvement in the vehicle technology (“vehicle-related measures”, as stated by the authors). But such measures (improved engine technology and emission standards) do aim to improve the emissions of multiple pollutants (NOx, THC, CO). This would not only reduce the emission of NOx, but also the emissions of ozone precursors.
  12. Line 97-98: Please cite those “few studies”.
  13. Line 112: Please delete the word “dangerous”. It’s unclear and irrelevant.
  14. Line 113: Replace the term “fresh air emitted by vehicles” with “fresh vehicular exhaust”.
  15. Figure 1: Not quite useful for readers; I would suggest moving it to the supplementary material.
  16. Line 131-143: The description of the sampling/analytical methods is inadequate. The authors must provide the details of each analytical instrument (brand, model, measurement parameters, measurement range, sensor types, accuracy, calibration, data recording frequency) used in this study.
  17. Line 152: The details of the analytical method have been referred to the author’s prior study. However, some details are still required in the manuscript. The authors should provide the QAQC parameters (e.g., linearity, recovery, MDL) of their analytical methods.
  18. Line 153-169: How were the samples of NOx, SO2, and NH3 collected? Did the authors used adsorbents? Where did the authors adopt these methods from (cite the sources)? The analytical methods and QA/QC of the methods are completely missing. E.g., in Line 158, the authors say- “…analyzed by a spectrophotometer”; what standard was used? What concentrations of the standards? What’s the linearity, recovery, accuracy, MDL of the methods? Please provide these details for each analytical method.
  19. Line 153-169: What types of filters were used? What was the sampling instrument for NOx, SO2, and NH3? Did the authors analyze the samples in the laboratory in Nepal? If the samples were shipped abroad for the analysis, how were the samples preserved? Did the authors collect these samples simultaneously?
  20. Line 167: How was NH3 collected on filters?
  21. Line 177-180: Please provide the values of all coefficients, and also mention how were those values obtained.
  22. Line 180-181: The background levels are adapted from a study conducted in China. To what extent would those values represent background concentrations in the context of the author’s study area?
  23. Section 2 (methods): Please provide the information on sample size. I also suggest the authors provide a photograph of the sampling setup in the supplementary material.
  24. Line 200-208: Please indicate what measure does each value represent. Mean or single measurement value or what? Provide standard deviation or interquartile range or other suitable measures of dispersion.
  25. Figure 2: The VOC concentration increased from 9 am to 11:30 am at Koteshwor, but it decreased during the same duration at Sanepa. Please discuss the possible reasons for this.
  26. Line 224: How many observations were used to calculate the means? As I mentioned above, the sample size is unclear. Please provide the uncertainty measures with the mean values.
  27. Line 249 and Figure 4: Which pollutant is discussed here? The authors did not mention the pollutant. E.g., “the contribution of gasoline was approximately 90%”, but to what?
  28. Table 2. Please indicate the number of observations used for the calculation of correlation coefficients.
  29. Section 3.3. There exist several more advanced tools of source apportionment. The correlation analysis presented here is overly simplistic. The results also lack the analysis of uncertainty. Moreover, the analysis presented here treats as if the samples were purely contributed by the traffic emission sources. But the samples, in fact, were affected by multiple sources.
  30. Line 294-295: “It was deemed that diesel exhaust emissions or resuspended dust had a significant impact on PM2.5 levels.” Where is the data to support this statement?
  31. Line 310-312: The terms like “significantly different” should be used only with appropriate statistical analyses. On what basis do the authors claim that “levels at Koteshwor were not significantly different from those in 2003 (Line 310)”? Their data (Table 3) show that most VOC concentrations were more than double the concentrations cited from the source Kondo (2003). Compare the concentrations of propane, propylene, isobutene, n-butane, for example. The authors’ claim is misleading and not supported by their data.
  32. Table 3: The citation format is different from that in the rest of the manuscript, which makes it difficult to locate the cited sources in the reference list. The cited sources Sharma (1998) and Kondo (2003) are not listed in the reference list (references with different publication years are listed in the reference; please check it).
  33. Line 321-322: “It was shown that aromatics and alkenes significantly contributed to ozone production through long-distance and short-distance transport, respectively.” This statement is not supported by any data presented by the authors.
  34. Line 330-332: Where is the correlation analysis discussed here? Where are the supporting data?

Round 2

Reviewer 3 Report

The authors have addressed most of my comments. However, there are still some issues that need to be addressed before publication. My comments and suggestions are as follows:

  1. Line 129: Please check the spelling “Suuma”. Should it rather be “Summa”?
  2. Line 158: What is the meaning of “accuracy of ±n° and ±an”? Did the authors intend to present the actual value (instead of n) here? Please correct.
  3. Line 164: “kept cool” is not clear. Please mention the temperature.
  4. Table 2: Are the standard deviation values calculated for the samples from Koteshwor and Sanepa combined? Please specify it (a footnote can be added), and also mention the sample size (n=?) in the SD column heading.
  5. Table 4 and related discussions: The sample size (n = 4) is too small for the correlation analysis. Also note that some compound concentrations were below detection limits (Table 2). In that case, the available sample number for the correlation analysis is even smaller. Therefore, the authors should indicate the limitation of their analysis in the related discussion.
  6. Table 3, last 3 rows: Do these values represent raw exhaust concentration? It seems like the values are adopted from prior studies (because the methods section does not describe about exhaust sampling). Please indicate the data source.
  7. Line 391-392: “The ratios of gasoline burned components to that of gasoline unburned components in this study has been greater than that in 1998.” What are those ratios? And where did the authors present the comparison of “ratios of burned to unburnt gasoline components” in the different years? Table 5 compares only the concentrations (ppb, not ratios). I did not find supporting data for this statement. Please provide the supporting data.
  8. Conclusion: Please mention the type of pollutant in Line 434-435 (because for some pollutants, e.g., particulate matter, DVs are more important than GVs). The authors concluded that gasoline vehicles are the major source of pollution in the Kathmandu valley based on a very small sample (n = 4 at a site). The authors should state the limitation of the study along with their conclusions to aware readers of it.
